# Clinical Characteristics of Minimal Lumbar Disc Herniation and Efficacy of Percutaneous Endoscopic Lumbar Discectomy via Transforaminal Approach: A Retrospective Study

**DOI:** 10.3390/jpm13030552

**Published:** 2023-03-20

**Authors:** Feifei Chen, Guihe Yang, Jinjin Wang, Zhongpeng Ge, Heran Wang, Yifei Guo, Heng Yang, Xingzhi Jing, Xiaoyang Liu, Xingang Cui

**Affiliations:** 1Department of Spine Surgery, Shandong Provincial Hospital Affiliated to Shandong First Medical University, Jinan 250021, China; 2Department of Spine Surgery, Shandong Provincial Hospital, Shandong University, Jinan 250021, China

**Keywords:** minimal, lumbar disc herniation, percutaneous endoscopic lumbar discectomy, minimally invasive treatment

## Abstract

Objective: To define the characteristics of Mini LDH, develop new diagnostic references and examine the clinical efficacy of percutaneous endoscopic lumbar discectomy via a transforaminal approach (TF-PELD) for it. Methods: A total of 72 patients who underwent TF-PELD with Mini LDH from September 2019 to October 2022 were enrolled in this retrospective study. The patients’ basic information, symptoms, number of outpatient visits, duration of conservative treatment, physical examination findings and so on were obtained from the medical records. Clinical effects of TF-PELD for Mini LDH were assessed by means of the following: the Visual Analog Scale (VAS) for low back pain (LBP) and leg pain, Oswestry Disability Index (ODI) for functional status assessment and Modified Mac Nab criteria for patient satisfaction. Results: Mini LDH have specific clinical characteristics and imaging features. All included patients achieved obvious pain relief after TF-PELD surgery. Pain scores were repeated at postoperative day 1 and 1, 3, 6, 12 and 24 months later. Results were statistically analyzed. The average VAS-Back, VAS-Leg and ODI scores were all significantly reduced at the first postoperative day and gradually decreased with the follow-up time continuing. In total, 66 out of 72 patients received an excellent or good recovery and no poor result was reported according to the Modified Mac Nab criteria. Conclusions: Mini LDH is a type of LDH with special characteristics and in need of correct diagnosis and active treatment in clinical work. TF-PELD was also found to be an effective procedure for the treatment of Mini LDH.

## 1. Introduction

Lumbar disc herniation (LDH) is a common back disorder, which is often manifested as a portion of the intervertebral disc breaking out to the surrounding space. The herniated tissues compress and irritate nearby nerves via inflammation, giving rise to low back pain (LBP) and/or typical sciatica, numbness and weakness of the lower extremities [1,2]. According to statistics, the overall prevalence of LDH is approximately 1% to 3% of the population in the United States and Europe [3,4], and 7.62% in a province of China [5,6]. Compared with common LDH, Minimal Lumbar Disc Herniation (Mini LDH) has a low morbidity and then is more likely to be frequently ignored and misdiagnosed in clinical work.

Mini LDH is a special kind of LDH. It does not have relatively remarkable disc herniation on Computed Tomography (CT) or Magnetic Resonance Imaging (MRI), but does have severe pain. It has already brought great challenges to accurately recognizing Mini LDH. Mini LDH, although uncommon, causes significant pain, discomfort and sometimes disability. However, few studies have systematically explored it in recent years. In the present study, we aim to define the characteristics of Mini LDH, develop new diagnostic references and examine the clinical efficacy of percutaneous endoscopic lumbar discectomy via s transforaminal approach (TF-PELD) for it.

## 2. Materials and Methods

### 2.1. Ethics Compliance

We conducted a clinical retrospective, observational study approved by the Institutional Review Board of our hospital (No. 2019-013, 18 April 2019). Due to the retrospective nature of this study, the informed consent was waived by the regional ethical review authority. We have, respectively, adhered to the Declaration of Helsinki and the Strengthening the Reporting of Observational studies in Epidemiology (STROBE) guidelines while conducting and reporting this study [7].

### 2.2. Patient Population

From September 2019 to October 2022, 1226 patients underwent TF-PELD for LDH in our hospital. Inclusion criteria were (1) patients who were diagnosed with LDH at one level with no prior or subsequent surgery at any other spinal level; (2) the size of LDH did not exceed 1/4 of anteriorposterior diameter of the spinal canal on CT/MRI (Figure 1). A total of 136 out of 1226 patients statistically met the inclusion criteria. We excluded patients from our study who met at least one of the following criteria: (1) patients who underwent multiple levels of discectomy, or concomitant surgery in addition to PELD performed at the same or different levels; (2) patients who underwent prior interventional pain procedures; (3) patients with stenosis, infection, fractures or tumors. Of these, 64 patients were excluded for meeting the exclusion criteria. Finally, a total of 72 patients were enrolled in this study (Figure 2). 

### 2.3. Medical History Collection and Radiologic Features

The patients’ basic information, symptoms, number of outpatient visits, duration of conservative treatment, physical examination findings and so on were obtained from the medical records and evaluated systematically. Evaluation of disc location, disc type, disc size on CT/MRI and calcification of the protruding intervertebral disc were blinded and performed by 3 surgeons. If more than 2 surgeons agreed, their recommendations were adopted and recorded.

### 2.4. Outcome Measurements and Follow-Up

The 10-point visual analogue scale (VAS) was used to assess back pain (VAS-Back) and leg pain (VAS-Leg). The Oswestry disability index (ODI) was adopted to evaluate patients’ function status. The VAS scores and ODI scores were collected and recorded at the following time points: pre-operation, 1st postoperative day, 1st postoperative month, 3rd postoperative month, 6th postoperative month, 12th postoperative month and 24th postoperative month. Modified Mac Nab criteria was used to analyze patients’ satisfaction at 24th postoperative month.

### 2.5. Surgical Procedure

Previous clinical studies have confirmed that, compared with the conventional surgery, TF-PELD has obvious advantages such as smaller incisions, faster recovery, decreased damage to soft tissues and fewer postoperative complications [8,9]. Additionally, patients with almost all types of LDH could be treated by TF-PELD with the advances in equipment and technique, allowing percutaneous approach of foraminoplasty [10]. In the treatment of LDH, more and more patients tended to choose TF-PELD first. 

Similarly, TF-PELD was also the first choice for the treatment of Mini LDH. Our surgical procedure to remove Mini LDH was based on the conventional endoscopic approach. The procedure was performed in the prone position on a radiolucent table under local anesthesia. No sedatives were used during the procedure. Before operating, the local anesthetic drugs were prepared from a mixture of 0.25% ropivacaine in 4 mL, 0.5% lidocaine in 10 mL and 0.9% normal saline in 16 mL. Under the guidance of G-arm fluoroscopy, the surgical segment and puncture needle entry point was determined. After infiltration of the entry point with 3 to 5 mL of the mixed narcotic drug, an 18-gauge spinal needle was introduced, slid to the facet joint and further advanced through the foramen to the target site of the nucleus pulposus protrusions cautiously under G-arm fluoroscopic guidance. During the puncture procedure, about 16 to 20 mL of the mixed narcotic drug was injected into the skin, subcutaneous tissue, fasciae, muscle and the lumbar facet joint layer-by-layer. The mixed narcotic drug was added during the operation, if necessary [11]. 

An incision of approximately 7 mm was made in the skin after a guide wire was advanced through the 18-gauge needle. The dilator and cannula were subsequently inserted from external to internal layer by layer along the guide wire. After placement of the 7.5 mm working cannula, the herniated disc was removed using endoscopic forceps. Adequate exposure of the dura and nerve root, mobilization or pulsation of the neural tissues are the criteria for complete decompression. When the bleeding was stopped using bipolar probes, the endoscope and working cannula were removed and the incision was closed. Continuous feedback was obtained from the patient to prevent damage to neural structures during the entire procedure [12].

### 2.6. Statistical Analysis

In this study, the Visual Analog Scale score for the leg (VAS-L) was the most important observation indicator in this paper. According to the consulted literature [11,13], and the pre-observation results of clinical cases, the difference of the mean VAS-L scores was 1, i.e., *δ* = 1; the sample standard deviation is 1.12, i.e., *σ* = 1.12. Set bilateral *α* = 0.05, *Z_α_* (0.05) = 1.96; with 90% certainty, *Z_β_* (0.9) = 1.28. The following sample size calculation formula was used:$$n=\frac{{\left(Z_{\alpha }+Z_{\beta }\right)}^{2}*2\sigma^{2}}{\delta^{2}}$$

Considering the loss of follow-up rate and refusal rate of about 20%, 32 subjects at least were required. SPSS software (version 25.0, IBM Corp., Chicago, IL, USA) was used for statistical analysis. Continuous variables were presented as mean ± SD. To assess statistical significance of the continuous variables, such as VAS-Back, VAS-Leg and ODI scores, at different time points, student’s t-test was performed. The significance level was set to be 0.05.

## 3. Results

### 3.1. Patient Characteristics

Characteristics are listed in Table 1. A total of 32 males and 40 females were accepted for TF-PELD surgery for Mini LDH in these levels (L1/2, L2/3, L3/4, L4/5 and L5/S1). The median age of these included patients was 36 years old (range 18–72 years). Among these, sixty patients visited the outpatient clinic more than three times and the duration of conservative treatment was more than 3 months in fifty-two patients. Almost all included patients had positive straight leg raising (SLR) tests and normal heel and toe walks.

### 3.2. Imaging Features

The location of the herniated disc in relation to the pedicles and spinal canal was identified as central, paracentral, foraminal and extraforaminal herniation (Figure 3). The types of disc herniation were classified as shoulder and axillary based on the relationship with the dural sac and traversing nerve roots (Figure 4). According to the degree of upward and downward protrusion of the disc, it is divided into six types, namely, overly up-migrated, overly down-migrated, moderately up-migrated, moderately down-migrated, slightly up-migrated and slightly down-migrated (Figure 5). Imaging characteristics on CT/MRI are listed in Table 2. In accordance with the preoperative CT/MRI scans, we found the location of herniated disc in all included patients was paracentral and the integrity of the posterior longitudinal ligament (PLL) was compromised in almost 94.44% of them with high signal intensity on MRI T2W (Figure 6). The types of disc herniation were divided into shoulder in 60 patients and axillary in 12 patients. Nearly 83.33% of the included patients migrated slightly upward and downward. The sizes of the herniation were all less than 50% of the spinal canal compromise. Moreover, the disc herniation indices (DHIs) were all less than 1/4 and most (94.7%) of them were between 1/16 and 1/8 [14]. Less than 3.00% of subjects had calcification problems with protruding discs.

### 3.3. Clinical Outcomes

The herniated disc was completely removed in all cases. Significant pain relief was obtained for all enrolled patients after TF-PELD surgery. The average VAS-Back scores were reduced from preoperative 5.96 ± 2.10 to 3.37 ± 0.97 (*p* < 0.05) at the 1st postoperative day and reduced to 1.57 ± 1.01 (*p* < 0.05) at the 24th postoperative month. The average VAS-Leg scores were reduced from preoperative 7.83 ± 1.31 to 3.12 ± 1.00 (*p* < 0.05) at the 1st postoperative day and reduced to 1.17 ± 0.89 (*p* < 0.05) at the 24th postoperative month. As for function improvement, the average ODI scores improved from preoperative 47.60 ± 12.13 to 20.59 ± 11.43 (*p* < 0.05) at the 1st postoperative day and reduced to 11.00 ± 3.26 (*p* < 0.05) at the 24th postoperative month (Table 3). In total, 91.67% of patients received an excellent or good recovery and no poor result was reported by the Modified Mac Nab criteria (Table 4).

### 3.4. Case Presentation

A 38-year-old female patient came to the spinal clinic for severe left leg radicular pain of nearly 3 months. This patient was diagnosed with Mini LDH in L4/5 level on the basis of the physical examination and imaging findings. The herniated disc was completely removed during the surgery. This patient received immediate pain relief and was discharged at the first postoperative day. During the follow-up for 24 months, the functional improvement was satisfactory (Figure 7).

## 4. Discussion

LDH is one of the disturbing disorders which presents with the main symptoms of LBP and sciatica and imposes a heavy economic burden on people, families and countries [15]. It is reported that the incidence rate of LDH is as high as 20–35% among people over 50 years old [16], whereas Mini LDH is an unusual type of LDH. Because of its rarity and unique clinical manifestations, Mini LDH is usually misdiagnosed, or the diagnosis is delayed. 

In this study, we found that compared with common LDH, Mini LDH has the following clinical characteristics: (1) acute onset, generally less than 1 month; (2) on CT/MRI axial scan, the location of LDH was paracentral, in which the herniated disc can directly compress and irritate nearby nerves; (3) the integrity of the PLL is compromised with high signal intensity on MRI T2W; (4) the intervertebral disc, herniated through a tear in the PLL, can be observed under an endoscopic view; (5) severe pain, even at least VAS-Back score > 6.00 and/or VAS-Leg score > 8.00; (6) the size of LDH was small, generally DHIs < 1/8; (7) the disc usually migrated slightly upward and downward; (8) calcification rarely occurs in protruding discs; (9) conservative treatment is often ineffective, even if it lasted no less than 3 months. So, we can assume that LDH with the above characteristics can be diagnosed as Mini LDH. 

Previous studies have suggested that conservative treatment is a primary choice for symptomatic patients and 90% of LDH cases could be resolved with conservative measures, including oral drug therapy, acupuncture, chiropractic, manipulation and so on [17,18,19,20]. Yu P et al. considered that conservative treatment may be a good option for patients who are reluctant to undergo surgery, or who have any contra-indication for surgery [21]. In 2014, Chiu et al. believed that most LDH can be absorbed spontaneously after conservative treatment. Because penetrating the annulus fibrosus and the PLL result in being exposed to the systemic circulation in the epidural space, trusion and sequestration LDH had higher regression rates than bulging and protrusion LDH [22,23]. This may be the reasons why Mini LDH does not reabsorb well. Mao F et al. provided further evidence that LDH can be spontaneously absorbed without surgical treatment in 2022. Researchers deeply confirmed matrix metalloproteinases, macrophage regulation of inflammatory mediators and specific cytokines in intervertebral disc are essential for the spontaneous re-absorption of LDH [24]. 

However, there are many other studies that take different views. Sutheerayongprasert C et al. found that long-duration, sequestered herniation and large fragment are predictive of failure in the conservative treatment of LDH [25]. Gugliotta M et al. believed that surgical treatment provided faster pain relief in patients with LDH compared with conservative therapy [26]. In clinical work, the decision on whether or not to surgical intervention depends on the size of herniated disc. Generally speaking, the probability of surgery is greater when the herniated disc is larger. Some views also hold that the herniation causing nerve displacement was a prerequisite for being considered for this procedure. However, recently, Gupta et al. pointed out that it is not reasonable to use the size of the herniated disc (as a percentage of the spinal canal area) and nerve displacement to predict whether surgery was needed [27,28]. Similarly, we confirmed that the size of Mini LDH was exactly not large but nerve roots were precisely jammed. Severe pain is difficult to relieve in this state. We also found that the included patients with Mini LDH visited the outpatient clinic more than three times at least and were given conservative treatment, but had no diminishment of symptoms within 3 months. In view of the above situation, we deem that patients diagnosed with Mini LDH should undergo surgical intervention if their symptoms did not improve or even became worse after twelve weeks of conservative treatment [29].

Among spinal surgery technologies, TF-PELD has various advantages, including less bleeding, less trauma, less dissection of muscle tissue, preservation of bony structures and faster recovery [8,30,31,32]. In TF-PELD, surgeons use a unilateral single channel to directly reach the target position through Kambin‘s triangle, remove the herniated disc tissue and release nerve roots [33,34]. In this paper, all 72 patients received remarkable pain relief. The average VAS-Back, VAS-Leg and ODI scores were all significantly reduced at the first postoperative day and gradually decreased with the follow-up time continuing. In total, 66 out of 72 patients received an excellent or good recovery and no poor result was reported by the Modified Mac Nab criteria. Therefore, TF-PELD was an effective procedure for the treatment of Mini LDH.

There were some limitations to this retrospective study. Firstly, the sample size of this study was small, and the follow-up period was short. Secondly, this study was led by one experienced spine surgeon, and it was impossible to generalize his experience and these findings to all spine surgeons. Thirdly, our results are based on enrolled patients’ self-reported data. Finally, there is no comparison with other therapeutic options.

## 5. Conclusions

Mini LDH is a type of LDH with special characteristics, which was summarized systematically in this paper. Clinical characteristics of Mini LDH would be very helpful to provide a definitive diagnosis for pain physicians. Conservative treatment was often useless while TF-PELD was found to be an effective procedure for the treatment of Mini LDH due to significant postoperative pain relief. Future studies are needed in order to evaluate the effectiveness of TF-PELD for Mini LDH in a longer follow-up time.

## Figures and Tables

**Figure 1 jpm-13-00552-f001:**
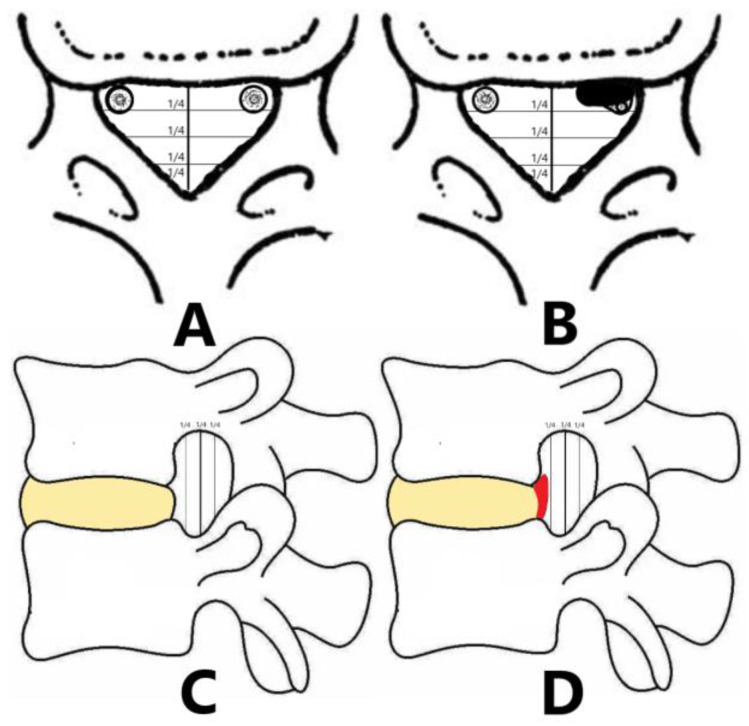
The proportional relationship between LDH size and spinal canal: (**A**,**B**) Imaging findings of Mini LDH in axial view. (**C**,**D**) Imaging findings of Mini LDH in sagittal view.

**Figure 2 jpm-13-00552-f002:**
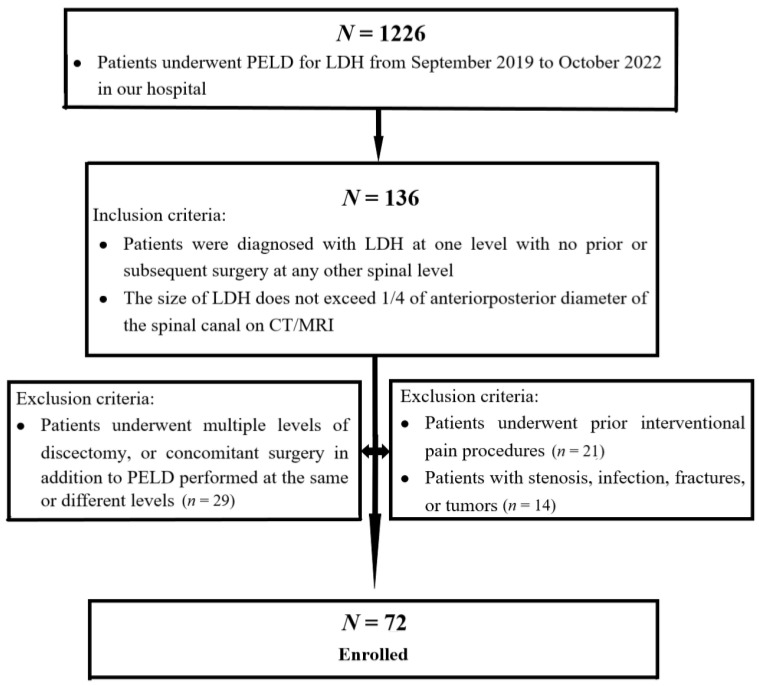
Flow diagram of patients enrolled in this study.

**Figure 3 jpm-13-00552-f003:**
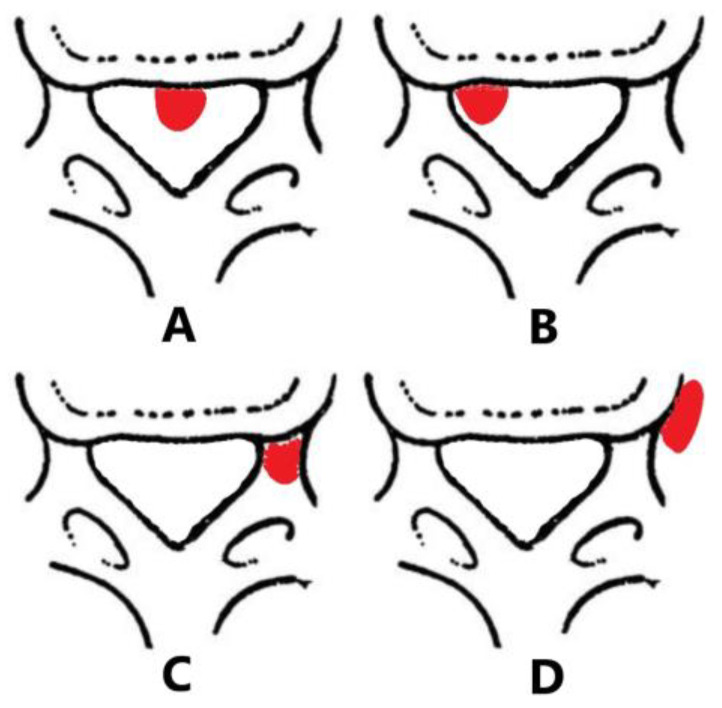
The location of LDH: (**A**) Central. (**B**) Paracentral. (**C**) Foraminal. (**D**) Extraforaminal.

**Figure 4 jpm-13-00552-f004:**
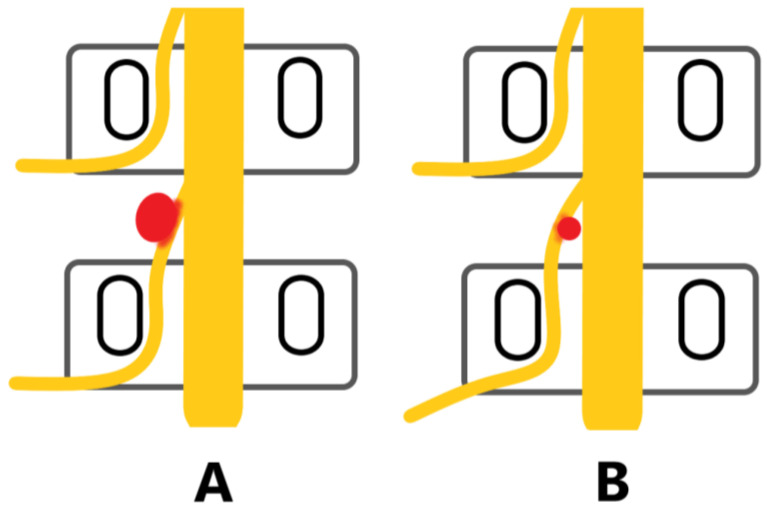
The type of the herniation: (**A**) Shoulder. (**B**) Axillary.

**Figure 5 jpm-13-00552-f005:**
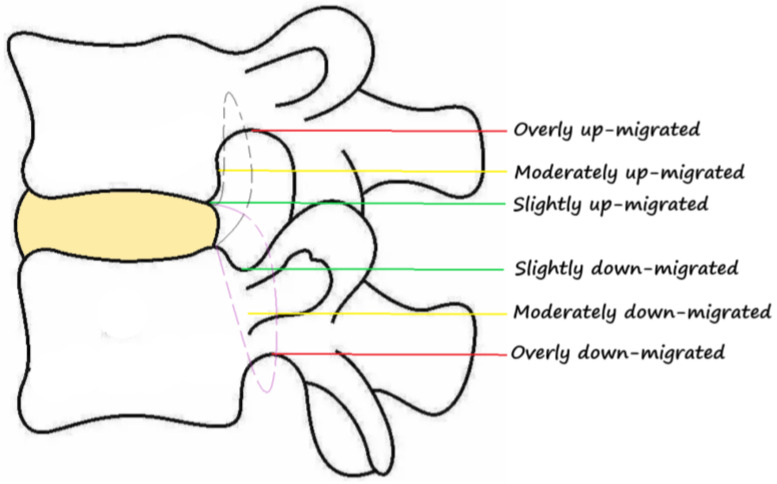
The degree of upward and downward protrusion of the disc.

**Figure 6 jpm-13-00552-f006:**
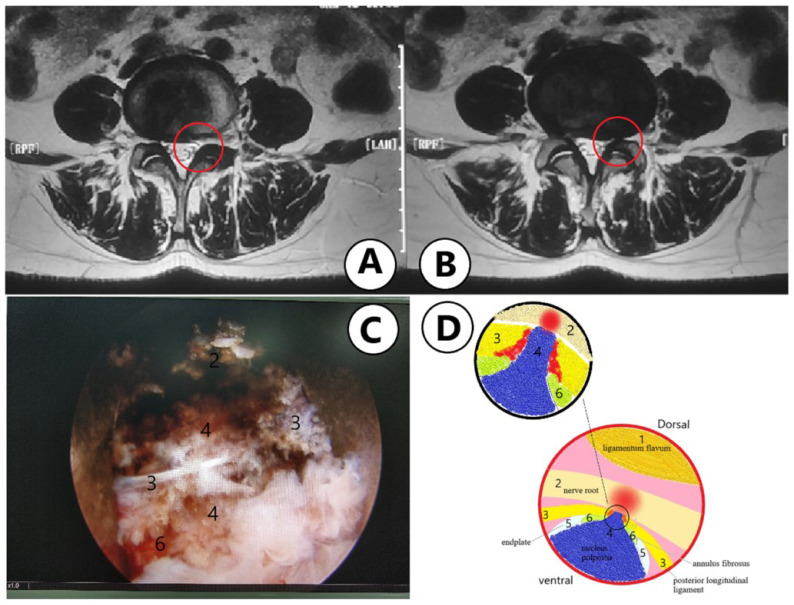
Imaging features of Mini LDH: (**A**) Paracentral herniation on MRI axial scan. (**B**) The PLL is compromised with high signal intensity on T2W. (**C**) Mini LDH in PELD. (**D**) Schematic diagram of Mini LDH in PELD.

**Figure 7 jpm-13-00552-f007:**
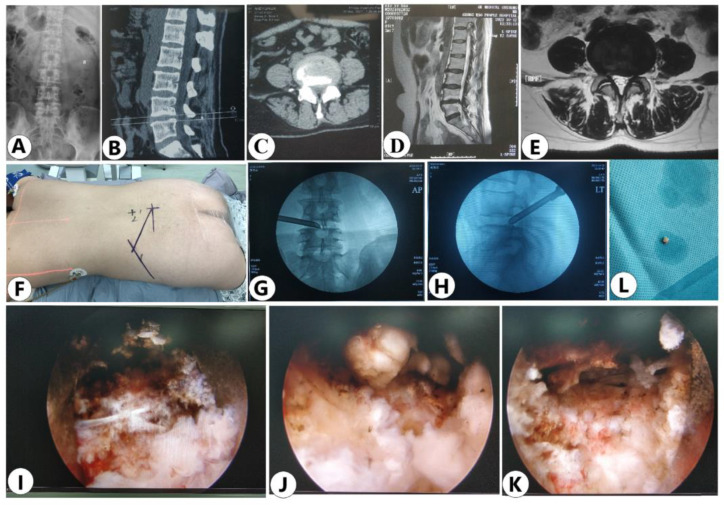
Case presentation: (**A**) Anterior-posterior X-ray image of lumbar spine. (**B**,**C**) CT sagittal and axial view of L4/5. (**D**,**E**) MRI sagittal and axial view of L4/5. (**F**) The rough shapes of anatomical structures such as pedicles and iliac crest were drawn on skin, and puncture targets and trajectories were marked on skin. (**G**) Anteroposterior X-ray view of the working cannulas at L4/5. (**H**) Lateral X-ray view of the working cannulas at L4/5. (**I**) The PLL is compromised before discectomy under endoscopic view. (**J**) The protruded nucleus pulposus under endoscopic view. (**K**) The decompressed nerve root under endoscopic view. (**L**) Small migrated fragments were removed.

**Table 1 jpm-13-00552-t001:** Characteristics of the included patients.

Item		Average (%)
Total Number		72
Gender	Male	32 (44.44%)
	Female	40 (55.56%)
Segments	L1/2	2 (2.78%)
	L2/3	3 (4.17%)
	L3/4	7 (9.72%)
	L4/5	32 (44.44%)
	L5-S1	28 (38.89%)
OP method	TF-PELD	72 (100%)
Side	Left	34 (47.22%)
	Right	38 (52.78%)
Age	<30 years	10 (13.89%)
	30–40 years	36 (50.00%)
	40–50 years	14 (19.44%)
	>50 years	12 (16.67%)
Number of outpatient visits	<3	12 (16.67%)
	≥3	60 (83.33%)
Duration of onset	<1 month	67(93.06%)
	≥1 months	5(6.94%)
Speed of onset	Acute	69(95.83%)
	Chronic	3(4.17%)
Duration of conservative treatment	<1 month	3 (4.17%)
	(1 month, 3 months)	17(23.61%)
	≥3 months	52 (72.22%)
Straight leg raising test	Positive	70 (97.22%)
	Negative	2 (2.78%)
Heel and toe walk	Normal	69 (95.83%)
	Impaired	3 (4.17%)

**Table 2 jpm-13-00552-t002:** Image characteristics on CT/MRI.

Item		Average (%)
Total Number		72
The integrity of PLL	YES	68 (94.44%)
	NO	4 (5.56%)
Disc Location	Central	0 (0.00%)
	Paracentral	72 (100.0%)
	Foraminal	0 (0.00%)
	Extraforaminal	0 (0.00%)
Disc Type	Shoulder	60 (83.33%)
	Axillary	12 (16.67%)
Disc Size	≥50% canal compromise	0 (0.00%)
	<50% canal compromise	72 (100.0%)
Migration	Overly up-migrated	0 (0.00%)
	Overly down-migrated	0 (0.00%)
	Moderately up-migrated	5(6.94%)
	Moderately down-migrated	7(9.72%)
	Slightly up-migrated	33 (45.83%)
	Slightly down-migrated	27(37.50%)
DHI	<1/16	4 (5.56%)
	[1/16, 1/8)	66 (91.67%)
	[1/8, 1/4)	2 (2.78%)
	[1/4, 1/2)	0 (0.00%)
	≥1/2	0 (0.00%)
Calcification	YES	2 (2.78%)
	NO	70 (97.22%)

PLL: posterior longitudinal ligament; DHI: disc herniation index.

**Table 3 jpm-13-00552-t003:** Pain relief and functional improvement.

Variables	*N*	VAS-Back		VAS-Leg		ODI	
		Mean ± SD	Median (Min-Max)	Mean ± SD	Median (Min-Max)	Mean ± SD	Median (Min-Max)
Preoperative	72	5.96 ± 2.10	6.00 (1.00–10.00)	7.83 ± 1.31	8.00 (1.00–10.00)	47.60 ± 12.13	50.00 (10.00–90.00)
Postop. 1st day	72	3.37 ± 0.97 *	3.00 (1.00–8.00)	3.12 ± 1.00 *	3.00 (1.00–6.00)	20.59 ± 11.43 *	30.00 (5.00–55.00)
Postop. 1st month	72	3.01 ± 1.07	3.00 (1.00–8.00)	2.74 ± 1.20	3.00 (1.00–6.00)	20.04 ± 10.43	20.00 (5.00–45.00)
Postop. 3rd month	72	2.63 ± 1.08	1.00 (1.00–5.00)	2.00 ± 0.97	2.00 (1.00–4.00)	19.64 ± 9.87	20.00 (5.00–40.00)
Postop. 6th month	72	2.40 ± 0.86	1.00 (1.00–5.00)	1.49 ± 1.07	1.00 (1.00–3.00)	19.00 ± 7.05	20.00 (5.00–35.00)
Postop. 12th month	72	1.98 ± 1.01	1.00 (1.00–5.00)	1.23 ± 0.77	1.00 (1.00–3.00)	13.00 ± 3.01	13.00 (0.00–25.00)
Postop. 24th month	46	1.57 ± 1.01	1.00 (1.00–5.00)	1.17 ± 0.89	1.00 (1.00–3.00)	11.00 ± 3.26	11.00 (0.00–20.00)

* *p* < 0.05 represents a statistical difference from preoperative data. VAS: visual analogue scale; ODI: Oswestry disability index; SD, standard deviation; Postop., postoperative.

**Table 4 jpm-13-00552-t004:** Modified Mac Nab criteria.

Grade	Excellent	Good	Fair	Poor
Patients (*n*)	36	30	6	0
Percentage (%)	50.00	41.67	8.33	0.00

## Data Availability

The data that support the findings of this study are available from the corresponding author upon reasonable request.

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
