# Peer review of "Clinical Characteristics of Minimal Lumbar Disc Herniation and Efficacy of Percutaneous Endoscopic Lumbar Discectomy via Transforaminal Approach: A Retrospective Study"

_jpm, 2023, doi:10.3390/jpm13030552_

Round 1
Reviewer 1 Report
Overall, it is well-organized paper, and it is reasonable to draw conclusions. Furthermore, this issue is very hot and important to diagnosis LDH and treat with operation. However, it is necessary to introduce the inclusion criteria of mini LDH so that it can be easy to understand for mini LDH once again. Furthermore, a more detailed description of how the diagnosis of mini LDH and set the surgical indication from the study's results is needed.
Before the publication of this paper, a method of suspicious diagnosis for miniLDH and decision making of surgical procedures of miniLDH should be essentially described.
Reviewer 2 Report
All experienced spine surgeons have had to deal with the unknown of treating symptomatic Mini-Herniations. This paper could guide spine surgeons to some alternatives in the treatment
1) You should clarify if you use any sedation with local anesthesia.
2) The reasons for carrying out surgery before three months in 20 cases.
3) It would be relevant to mention whether they have calcified LDH.
4) The Follow up is too short
Reviewer 3 Report
1. Line 94 mention the word mixed narcotic. Do the authors mean local anesthetics
2. line 102 says superior vertebral notch. Do you mean upper part of the neural foramen
3. The authors did not clarify if the procedure is only for minimal paracentral disc herniation.
4. The authors did not clarify is the shoulder or the axilla type of hernination makes any diffrence in patient selection ?
5. The authors did not mention if the hernination causing nerve displacement is a prerequiste for being considered for this procedure selection.
6. The approach is very lateral and the guage of the cannula is not mentioned. If the guage is not that different from regular minimal discectomy approach, then trauma to tissue may not be that diffrence.
7. was any neuromonitoring done ?
8. G arm or C-arm for flouroscopy ?
Thanks
